# Characteristics and Clinical Management Strategy of Petrous Apex Cholesterol Granulomas

**DOI:** 10.3390/cancers15041313

**Published:** 2023-02-18

**Authors:** Sanne de Bock, Walter Szweryn, Thijs Jansen, Josje Otten, Jef Mulder, Jérôme Waterval, Yasin Temel, Stijn Bekkers, Henricus Kunst

**Affiliations:** 1Department of Otorhinolaryngology and Head & Neck Surgery, Radboud University Medical Center, 6525 GA Nijmegen, The Netherlands; 2Dutch Academic Alliance Skull Base Pathology, Radboud University Medical Center and Maastricht University Medical Center+, 6229 HX Maastricht, The Netherlands; 3Department of Otology and Head & Neck Surgery, Maastricht University Medical Center+, 6229 HX Maastricht, The Netherlands; 4Rare Cancers, Radboud Institute for Health Sciences, Geert Grooteplein 21, 6525 EZ Nijmegen, The Netherlands; 5Department of Neurosurgery, Maastricht University Medical Center+, 6229 HX Maastricht, The Netherlands

**Keywords:** petrous apex cholesterol granuloma, characteristics, wait-and-scan, surgery

## Abstract

**Simple Summary:**

Cholesterol granulomas are cystic lesions that can occur in the temporal bone. When such a lesion is present in the petrous apex, various clinical symptoms can occur due to compression of cranial nerves. Growth is observed in a minority of cases. Therefore, sole wait-and-scan management with intermittent magnetic resonance imaging during follow-up can be sufficient. Surgical drainage of the lesion is considered in case of severe symptomatology or growth. The aim of this consecutive case series was to evaluate the clinical characteristics of petrous apex cholesterol granulomas and outcomes after different management strategies. We confirm that PACGs are slow-growing lesions with a low risk of adverse events. Solely using the wait-and-scan strategy is a safe option for patients without symptoms, with acceptable symptoms and without symptom progression, and with asymptomatic growth. Surgical treatment can be considered in patients with symptom progression or symptomatic growth.

**Abstract:**

**Purpose:** To evaluate the clinical characteristics of petrous apex cholesterol granulomas (PACG) and assess outcomes after different treatment strategies. **Method:** A consecutive case series of 34 patients with a PACG. Main outcomes were PACG growth, symptoms, and the outcomes of different treatment strategies: wait-and-scan (WS) and surgical drainage. **Results:** Thirty-four patients were analyzed; mean follow-up time was 7.1 years. Twenty-one patients (61.7%) showed symptoms, mostly more than one. Most symptoms reported were cranial nerve palsy (58.8%) and headache (35.3%). Twenty-one patients (61.8%) received solely wait-and-scan (WS), and thirteen patients (38.2%) underwent surgery, five of whom (38.5%) after an initial WS period. In the solely WS group, one (4.8%) developed new symptoms, and two (9.5%) reported symptom progression despite a stable granuloma size. Two (9.5%) showed granuloma growth on follow-up scans without symptom progression. Surgery consisted of drainage. Eleven (84.6%) of these thirteen patients reported partial recovery; one (7.7%) reported no recovery; and one (7.7%) reported full recovery of reported symptoms related to PACG. Among the patients with cranial nerve involvement, 7.7% showed full recovery after surgery; 84.6% showed partial recovery; and 7.7% did not recover. Adverse events occurred in five out of 13 patients who underwent surgery, all with full recovery. **Conclusions:** This study confirms that PACG are slow-growing lesions with a low risk of adverse events. Solely using wait-and-scan strategy is a safe option for patients without symptoms, with acceptable symptoms without symptom progression, and with asymptomatic growth. Surgical treatment can be considered in patients with symptom progression or symptomatic growth.

## 1. Introduction

Cholesterol granulomas (CG) are cystic lesions consisting of a fibrous capsule with a sterile content of cholesterol crystals, foreign body giant cells, and chronic inflammation [1,2]. In the temporal bone, CG mostly occur in the mastoid and petrous apex [1,2,3]. With an incidence of 0.6 per million in the general population, CG of the PA (PACG) form a rare clinical condition with pathophysiology yet to be defined [4,5]. The current hypothesis is that CG are caused by hemorrhage from exposed bone marrow in the cellular tracts of the petrous apex which coagulates and occludes outflow pathways. The sustained hemorrhage subsequently causes cyst expansion with bone remodeling and further exposure of marrow. Cholesterol is liberated from red blood cells via a process of anaerobic breakdown [2,6].

CG in the petrous apex can cause various clinical symptoms due to the compression of different cranial nerves [5]. The most common complaints are hearing loss, tinnitus, vertigo, and headache [7,8,9,10,11]. The lesion is typically hyperintense on T1- and T2-weighted sequences and has a no-diffusion restriction on diffusion-weighted sequences [1,5,12].

The reported incidence of lesion growth is low (3.7–14.3%). PACG treatment can comprise solely wait-and-scan (WS), while surgery can be considered in case of growth or symptoms related to the CG [1,3,9]. Surgical intervention includes drainage to the middle ear, mastoid cavity, or sphenoid sinus [3]. Drainage is achieved via a transmastoid approach, hypotympanotomy, or endonasal approach, dependent on CG location, the patient’s hearing status, and other critical structures involved [1,3,5,13]. A stent is commonly left in situ to facilitate drainage, promote aeration, and prevent recurrence [5,14]. Large lesions which extend intracranially are often drained via a middle fossa approach [3,5,15].

We aim to evaluate the characteristics of PACG and assess outcomes after different treatment strategies in a multi-centered case series. 

## 2. Materials and Methods

### 2.1. Study Design

This study is a case series of 34 patients diagnosed with PACG in the Radboud University (*n* = 24) and Maastricht University Medical Centers (*n* = 10). The objective was to explore the patients’ characteristics, PACG related symptoms and growth, and the outcomes of different (surgical) approaches. Data were obtained from medical records. Patients were acquired via a web-based application (CT CUE) that searched medical records for keywords in radiological reports. 

### 2.2. Participants

All patients were diagnosed with a PACG between January 1991 and March 2019. Inclusion criteria were PACG confirmed by MRI and a minimum follow-up duration of six months including CT and/or MRI scan (Figure 1). Patient demographics and characteristics are portrayed in Table 1.

This research was performed according to the national and international ethical standards and was approved by the regional committee on Research Involving Human Subjects. 

### 2.3. Primary Outcome Measure

Our primary goal was to assess the proportion of patients that showed PACG growth during follow-up. Growth was determined as an increase in diameter in any plane on CT or MRI, measured by an experienced neuroradiologist. Significant PACG growth was defined by an increase of 2 mm or more in any direction [1,3].

**Figure 1 cancers-15-01313-f001:**
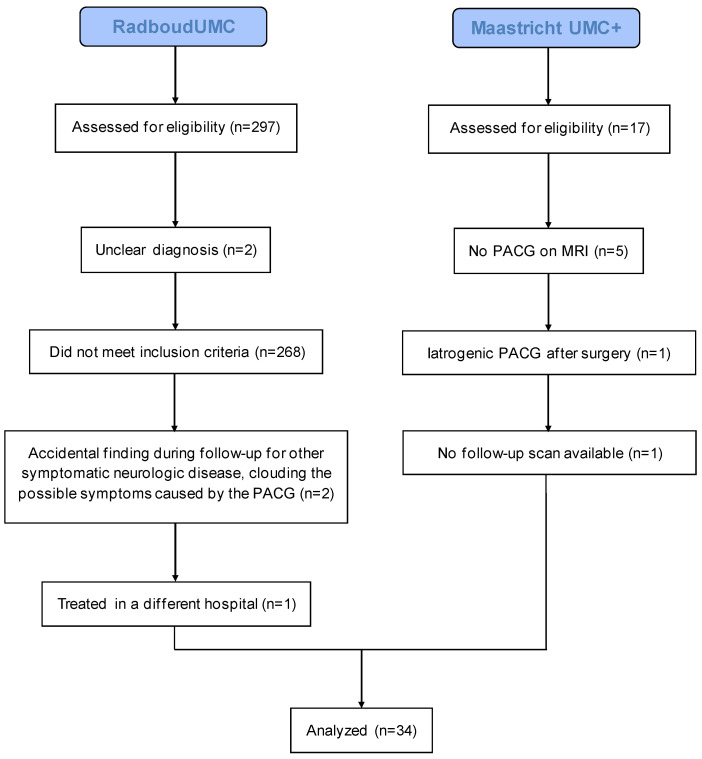
Flow diagram of recruited patients.

**Table 1 cancers-15-01313-t001:** Patient characteristics in current study.

Characteristic		Patients (*n* = 34)
Age in years at time of diagnosis (mean, range)		41.9, 8–77
Follow-up duration in years (mean, range)		7.1, 0.5–28.9
Sex, *n* (%)	Male	18 (52.9)
	Female	16 (47.1)
Location granuloma, *n* (%)	Left	17 (50)
	Right	18 (50)
Granuloma size in millimeters (mean, range)		22.1, 12–36
Reason for diagnosis, *n* (%)	Symptoms	18 (52.9)
	Coincidental	16 (47.1)

### 2.4. Secondary Outcome Measures

The secondary outcomes of the study were the following:The incidence and type of symptoms presumably related to a PACG included tinnitus, vertigo, hearing loss, and cranial nerve (CN) III, IV, V, VI, VII and X involvement;Concerning the different surgical interventions:
-The reason for surgical intervention: symptoms and/or growth;-The proportion of symptom relief (defined as full, partial, or no recovery), as observed by the clinician or reported by patient and clinician, and treatment success (absence of PACG recurrence on all follow-up scans) per surgical intervention;-The proportion of adverse events (temporary or permanent damage that cannot be considered normal perioperative course) per surgical intervention;-In the case of CN involvement before surgical intervention, the proportion of functional recovery post-surgery when compared to the pre-surgical condition.Concerning the number of patients treated with a WS strategy:
-The proportion of patients who developed new symptoms or symptom progression during the WS management, objectified by their clinician.

### 2.5. Statistical Analysis

Statistical analysis was performed using SPSS Statistics for Windows, version 25 (IBM, Chicago, IL, USA). Descriptive statistics were used to analyze the primary and secondary outcome measures. 

## 3. Results

Thirty-four patients were included in the analysis. Mean age at diagnosis was 41.9 years (range 8–77), and 47.1% of patients were female. Granuloma size at the time of diagnosis was 22.1 mm on average (range 12.0–36.0). The diagnosis was incidental in 47.1% of the patients. Eleven patients (32.4%) were lost to follow-up. Twenty-three of 34 patients (67.6%) still received follow-up at the time of data collection, including 13 in the WS group and 10 in the surgery group. Five of 26 (19.2%) patients who initially adhered to solely WS were eventually operated on because of granuloma growth or symptom severity.

### 3.1. Primary Outcome: Granuloma Growth

Data on granuloma growth are depicted in Figure 2. Significant growth was observed in five (14.7%) patients during follow up, which occurred during a mean follow-up period of 3.5 years (range 0.4–14.0). Growth ranged from 4.0 mm to 14.0 mm. Among the patients that demonstrated growth, two (40.0%) received solely WS, and three (60.0%) underwent surgery. 

One patient showed gradual growth from the moment of diagnosis onwards during 14 years of WS, after which the PACG was drained. The other four patients who showed growth first went through a stable period without growth for a mean of 3.9 years (range 0.4–12.0). 

### 3.2. Secondary Outcomes: Symptoms

CN palsy (overall 58.8%) and headache were the most common symptoms (35.3%). Results are depicted in Table 2. 

**Figure 2 cancers-15-01313-f002:**
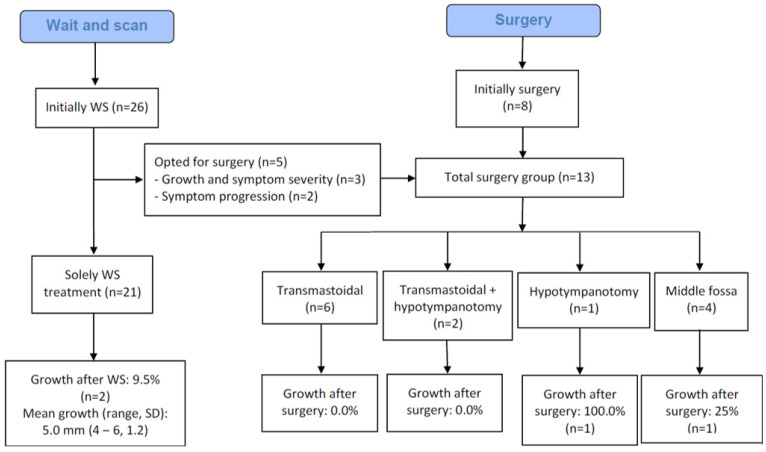
Routing of patients with outcome.

**Table 2 cancers-15-01313-t002:** Reported symptoms and clinical features.

Symptoms, *n* (%)		Yes	21 (61.8)
		No	13 (38.2)
Type of symptoms		*n* (% of total *n* = 34)	
CN involvement		20 (58.8)	
Abducens nerve palsy + facial numbness on the affected side		1 (2.9)	
Numbness face affected side		2 (5.8)	
Abducens nerve palsy		3 (8.8)	
Facial nerve spasms		3 (8.8)	
Oculomotor nerve palsy		1 (2.9)	
Trochlear nerve palsy		1 (2.9)	
Vagus and accessory nerve palsy		1 (2.9)	
Facial pain (trigeminal nerve involvement)		1 (2.9)	
Tinnitus		6 (17.6)	
Vertigo		9 (26.5)	
Hearing loss	Perceptive	1 (2.9)	
	Conductive	0 (0.0)	
	Mixed	4 (11.8)	
Headache		12 (35.3)	
Ear pressure		1 (2.9)	
Progression of symptoms, *n* (%)		Yes	8 (23.5)
		No	26 (76.5)

### 3.3. Secondary Outcomes: Treatment

Of the 21 patients who suffered from symptoms, 12 (57.1%) received surgery. One patient received surgery due to growth without symptoms. In total, 13 patients (38.2%) were operated on. Mean age at surgery was 45.4 years (range 33–69). The different approaches that were used were transmastoidal drainage (*n* = 8, 61.5%, in two patients combined with hypotympanotomy), hypotympanotomy alone (*n* = 1, 7.7%), and drainage via a middle fossa approach (*n* = 4, 30.8%). Drains were left permanently in situ to drain in the mastoid cavity in nine out of the twelve patients (75.0%). For routing of our patients, see Figure 2. A representative case is presented for demonstration purposes in Figure 3.

### 3.4. Secondary Outcomes: Surgical Outcomes

Mean duration of follow-up in the group that underwent surgery was 8.1 years (range 0.7–18.8).

### 3.5. Symptom Relief and Treatment Success

**Transmastoidal (*n* = 6).** The mean follow-up time in this group was 3.8 years (SD 5.2, range 0.69–14.2). One (16.7%) of six drained patients experienced no symptom recovery, and five (83.3%) reported partial recovery. Treatment was successful (no PACG recurrence or growth on all follow-up scans) in four (66.7%) patients. At time of data collection, treatment success was unknown in two patients; given their recent surgery, no follow-up scan had yet been made. Their first scans were made during the writing of this report: neither of them showed recurrence. These data were, however, not included in our analysis.

**Hypotympanotomy (*n* = 1).** A drain was left in situ after the procedure. The patient had a follow-up time of 18.8 years and reported partial symptom recovery. The treatment was unsuccessful: recurrence was reported after four years with stable symptomatology (Figure 2).

**Hypotympanotomy combined with a transmastoidal approach (*n* = 2).** One (50%) of these patients received a drain after surgery. There was a mean follow-up time of 6.3 years (SD 5.1, range 2.6–9.9). Both patients reported partial symptom recovery, and both treatments were successful (Figure 2).

**Middle fossa with concurrent drain placement (*n* = 4).** This group had a mean follow-up duration of 12.8 years (range 10.0–16.0). 

Three patients (75%) showed partial symptom relief, and one (25%) reported full recovery. After surgery with concurrent drain placement, no growth was reported in three patients. One patient showed recurrence after four years without symptomatology.

In one case, the approach was without drain placement, and after six years, growth was observed. With new insights, concurrent drain placement was performed with no reported recurrence during further follow up of another six years. 

All of the cases were presented in our multidisciplinary skull-base team before opting for surgery. An endoscopic approach was discussed in all of our cases but, due to unfavorable anatomy for endonasal drainage, a lateral approach was chosen. In retrospect—surprisingly—no endonasal route was chosen. Endoscopic transnasal resection still remains an option in lesions located in the petrous apex in our referral centers.

### 3.6. Secondary Outcomes: Adverse Events

**Transmastoidal.** There were four adverse events (66.7%) among the six patients in whom a transmastoidal approach was exerted (Figure 2). Three (50.0%) of the patients with adverse events had a per-operative cerebrospinal fluid (CSF) leak that was closed during the intervention, and one patient (16.7%) suffered from a 20 dB progression of pre-existent mixed hearing loss. However, during seven months of follow-up, this patient’s audiogram recovered to pre-operative values and even showed improvement in the lower frequencies compared to the pre-operative situation. Therefore, no long-term adverse events were reported for this approach.

**Hypotympanotomy.** There were no adverse events.

**Hypotympanotomy and transmastoidal.** These patients did not report adverse events.

**Middle fossa.** Two patients (50.0%) showed temporary facial nerve palsy with full recovery. 

### 3.7. Secondary Outcomes: Cranial Nerve Recovery

Out of the total of 34 patients, 20 (58.8%) showed symptoms of CN involvement. All patients who underwent surgery (*n* = 13) showed CN involvement. After surgery, one patient (7.7%) reported full neural recovery, 10 patients (84.6%) reported partial recovery, and one patient (7.7%) reported no recovery. Three patients retained abducens nerve palsy; however, in two of these three patients, the diplopia did relieve greatly. The other patient was surgically treated by the ophthalmologist. Another patient with abducens nerve palsy reported temporary neural recovery at first but noticed an increase in diplopia later on without signs of PACG growth. One patient retained vagal nerve symptoms but experienced relief of the accessory nerve symptoms. One patient retained symptoms of hearing loss, although the audiogram showed improvement after surgery. 

Of the seven patients with cranial nerve involvement who were not operated on, one (14.3%) showed minimal granuloma growth with a maximum amount of 4 mm. This patient did, however, not report any symptom progression. Another non-operated and formerly asymptomatic patient developed a new symptom during 9.2 years follow-up. This consisted of diplopia caused by abducens nerve palsy and was treated by ophthalmologic surgery.

### 3.8. Secondary Outcomes: Wait-and-Scan

Twenty-one patients received a solely WS strategy with a mean follow-up period of 6.5 years (SD 6.6, range 0.5–28.9). The above-mentioned patient who developed abducens nerve palsy was the only one who reported new symptoms during follow-up (4.8%). Two other patients (9.5%) reported progression of existing symptoms during 1.4 and 14.3 years of follow-up. Neither of these two patients showed PACG growth on their scans, and the symptoms were not severe enough to opt for surgery. PACG growth was seen in two other patients (9.5%); these patients showed no symptom progression.

From the 26 patients who initially started on WS, five (19.2%) eventually underwent surgery. Three (60.0%) were operated on because of PACG growth combined with symptom severity, two (40.0%) because of symptom progression. Mean duration of WS after initial diagnosis in the eventually-operated group was 8.2 years (range 0.5–14). Of these five patients, one patient had recurrence after four years.

## 4. Discussion

The present study evaluated the characteristics of PACG and assessed outcomes of different (surgical) approaches in a multi-centered consecutive case series. For the majority of patients in this study, a WS strategy was chosen. Growth, symptom progression, and development of new symptoms rarely occurred in this group (9.5%, 9.5% and 4.8%, respectively). 

One of the key goals when choosing a WS strategy is to have the opportunity to perform surgery at the right moment. Two patients received WS despite showing growth, one because the PACG did not cause symptoms and one because the symptoms were mild (mild and stable hearing loss). Five patients initially treated with WS eventually had to undergo surgery due to a combination of symptom severity, symptom progression, and growth. From this result, we conclude that the primary decisive factor in the management of petrous apex cholesterol granuloma is symptomatology. Growth seems to be a less important factor, however, it should also be taken into account. 

When looking at the general characteristics and management of PACG, our findings correspond to previous literature: PACG are slow-growing, benign lesions that can be treated with WS when symptoms are minor or absent and there is no or asymptomatic growth. In this case, growth can be accepted [1,9,13,17].

A total of thirteen patients underwent surgery, all of them in the form of drainage either via a transmastoid approach (*n* = 6), a combination of transmastoid approach and hypotympanotomy (*n* = 2), hypotympanotomy alone (*n* = 1), or a middle fossa approach (*n* = 4). Only one patient (7.7%), drained via a middle fossa approach, reported full symptom recovery. However, the operation had to be reported as unsuccessful because of PACG growth during follow-up (Treatment success required no sign of PACG recurrence on all follow-up scans). Furthermore, eleven patients (84.6%) showed partial recovery, and one (7.7%) did not recover at all. The latter patient received drainage through a transmastoid approach. The results related to symptom relief seem in line with previous research on PACG drainage, which shows symptom relief ranging from 77.4% to 82.4% [3,10]. These studies investigated different drainage approaches, including the middle fossa and transmastoid approaches. They did not specify how many patients showed full or partial recovery. 

According to past research, hearing loss, tinnitus, vertigo, and headache are the most frequent clinical symptoms of PACG, and CN VII, V, and VI are most frequently involved [7,8,9,10,11]. Symptom severity was the main reason to opt for surgery, and all the patients who underwent surgery exhibited symptoms of CN involvement.

Of the thirteen operations, nine (69.2%) proved to be technically successful (absence of PACG regrowth on all follow-up scans). Two patients (15.4%) showed recurrence, of whom one was drained by a middle fossa approach and one by a hypotympanotomy. In two patients, no follow-up scan had yet been made at the time of data collection. Our study’s recurrence rate after surgery is in line with previous research on PACG drainage where a recurrence rate between 14.7% and 21.4% is mentioned [3]. However, given the limited number of patients in each of the subgroups, no preferable surgical approach can be formulated. Moreover, the best surgical approach for drainage should be made based on the patient’s anatomy, anatomical characteristics of the lesion, and surgeon experience.

All the adverse events that we encountered are known complications and have been previously reported after PACG surgery. None of the adverse events in this study had long term consequences. Our study’s most frequent adverse event was a temporary per-operative CSF leak (23.1%, all in the transmastoid drainage group). This percentage is higher than in previous literature where intraoperative CSF leak occurred in 9.7–13.3% of the patients [3,9]. There also was one patient with temporary deterioration of pre-existent hearing loss and two patients with temporary facial nerve palsy. The proportions of these adverse events in our study correspond to those of past literature, where 12–22% temporary facial nerve palsy and 3.6% postoperative hearing loss are mentioned [3,9,18]. 

The majority of patients with CN involvement reported partial symptom relief after surgery (84.6%). These findings on CN recovery are disproportionate to CN recovery reported in a previous study that reported normal function in 89% of patients. Only 8.3% of our patients reported full CN recovery [9,10]. In Italy, only 44.5% of patients received a wait-and-scan treatment in contrast to 61.7% in our cohort. More conservative treatment could possibly explain the difference in CN recovery in our institute. 

In the treatment of this rare pathology, an indication for follow up duration could be of use in the standardizing of care. In the literature, to our knowledge, no recommendations are formulated about the duration of follow up of PACG after diagnosis or after surgery. In our population, growth and development of symptoms in the WS group occurred after a maximum of 14 years after first diagnosis (growth with symptoms). Concerning the drainage group, we observed growth after a maximum of four years and development of new symptoms nine years after surgery. After analysis of the occurrence of growth, symptoms, and recurrence, we recommend a follow up duration of 10 years. In our opinion, a scan interval of one, two, four, seven and ten years after diagnosis or surgery is justified. When dismissing patients from further follow-up, it is advised to counsel patients that in case of development of new symptoms a renewed assessment should be made. Our recommendations for a clinical pathway can be found in the flowchart of Figure 4. 

Limitations of the present study include that we mainly used subjective measurements to investigate symptoms and symptom relief. More consequent pre- and postoperative tone and speech audiometry could have provided more objective insight into the effects of PACG on hearing and the relief of those symptoms after surgery.

The variability in follow-up time also limits this study regarding the wait-and-scan strategy and the evaluation of treatment success after surgery. Possible future recurrences could therefore be missed. However, the mean follow-up time was considerably longer compared to what is reported in the literature. Moreover, given the previously mentioned rarity of PACG, retrospective research on the lesions is the highest grade achievable.

**Figure 4 cancers-15-01313-f004:**
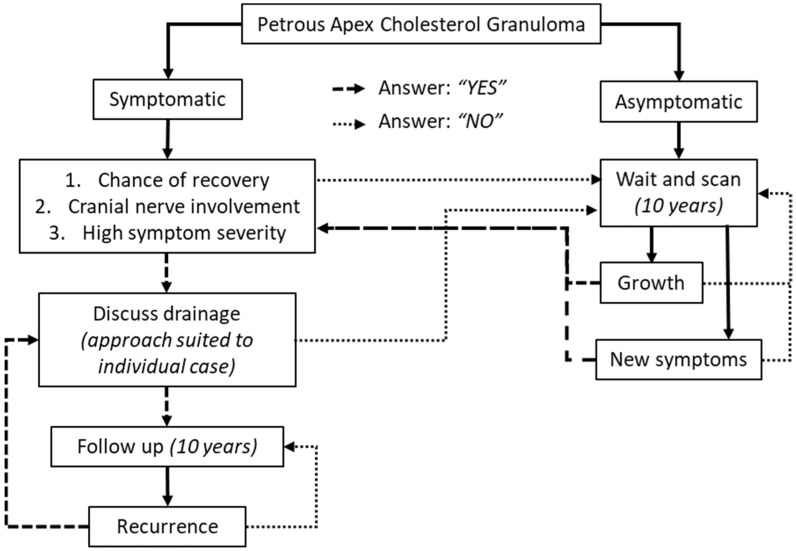
Clinical pathway when treating petrous apex cholesterol granulomas.

## 5. Conclusions

PACG are rare, slow-growing, and often asymptomatic benign lesions frequently found incidentally. Because of their growth characteristics and the low risk of adverse events, wait-and-scan treatment seems a safe option for patients without symptoms, with acceptable symptoms without symptom progression, and with asymptomatic growth. Surgical treatment may be indicated for patients with symptom progression or symptomatic growth. The vast majority of patients obtained symptom relief after surgery, and recurrence and adverse events related to surgery were limited. The surgical approach should be based on patient characteristics. A follow-up time of 10 years with MRI is advised after surgery and solely waiting-and-scanning.

## Figures and Tables

**Figure 3 cancers-15-01313-f003:**
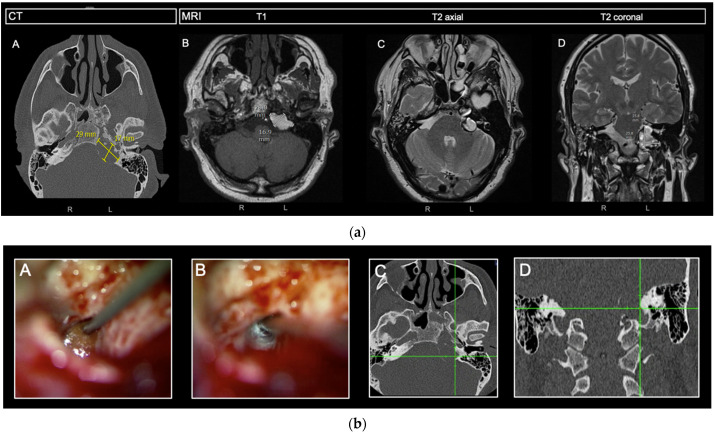
(**a**) Preoperative radiologic findings of a petrous apex cholesterol granuloma on the left side. (**A**) The axial CT reconstruction, used for navigation, of a low-density expansive mass in the left petrous apex with corresponding radiographic measurements. (**B**) T1 axial MRI reconstruction with corresponding radiographic measurements. (**C**,**D**) T2 axial and coronal MRI reconstruction. CT, computed tomography; MRI, magnetic resonance imaging. (**b**) Intraoperative findings with corresponding CT-navigation. (**A**,**C**) at the point of entry at the lateral wall of the PACG after mastoidectomy with subsequent drainage. (**B**,**D**) Identifying most medial wall of the PACG [16].

## Data Availability

Data is available on request via the authors.

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
