# Peer review of "Characteristics and Clinical Management Strategy of Petrous Apex Cholesterol Granulomas"

_cancers, 2023, doi:10.3390/cancers15041313_

Round 1

Reviewer 1 Report (New Reviewer)

I congratulate the authors for the good study presented in the article. Cholesterol granuloma is a rare pathology, with a relatively low number of published cases. This paper will undoubtedly help many readers to learn more about the disease and will be useful for managing their patients.

Author Response

We would like to thank the reviewer for thoroughly evaluating our work and seeing the relevance of the presented work

Reviewer 2 Report (New Reviewer)

This is an interesting article that shades lights on a rare skull base lesion. I have found it particularly valuable with regard highlighting the nature history of this disease ( petrous apex granuloma). However, I would recommend for the authors to provide figures that show and illustrate the radiological features of this disease ( pre op and post OP MRI and CT images)

Author Response

We would like to thank for the useful comment to improve the paper. Reviewer 3 also suggested adding intra-operative pictures. We have addressed this issue by adding both your suggestions to the article by means of an exemplary (transmastoidal) case (Figure 3-A and 3-B).

Reviewer 3 Report (New Reviewer)

Nice paper with relatively large number of a rare pathology. I think some improvements may improve this manuscript for publication:

  1. in the demographic table if they could add the exact locations of the granuloma instead of mentioning only the side 
  2. There are no radiographic or intraop figures. Perhaps a figure showing a collage of preop MRI axial view of the treated granulomas could give the reader a better idea of what type of lesions were treated? How about a view of the drain?
  3. There was no mention of an endoscopic approach. This was a bit surprising given that this has become a much more common approach to address PA cholesterol granulomas. I'm not sure if that is because of surgeon preference (perhaps more being seeing by neuro-otologists?). I would be curious to know more about approach selection. Where some of these possibly amenable to endonasal route but mastoid approaches chosen, or is this truly a cohort of laterally located petrous cholesterol granulomas, less so involving the petrous apex
  4. What was the recurrence rate and median follow-up time after surgery treatment?
  5. How long were drains left in place postop? where did the drains exit (i.e. where did the resection cavity contents drain out into?)

Author Response

We would like to say thank you to the reviewer for the useful comments to improve the paper. We have addressed all the comments as explained below.

  1. in the demographic table if they could add the exact locations of the granuloma instead of mentioning only the side

We have carefully selected a group of cholesterol granulomas limited to the apex of the petrous bone. Therefore no subdivision or subtyping of the lesion was performed.

  1. There are no radiographic or intraop figures. Perhaps a figure showing a collage of preop MRI axial view of the treated granulomas could give the reader a better idea of what type of lesions were treated? How about a view of the drain?

We have added your suggestions to the article by means of an exemplary (transmastoidal) case (figure 3-A and 3-B).

  1. There was no mention of an endoscopic approach. This was a bit surprising given that this has become a much more common approach to address PA cholesterol granulomas. I'm not sure if that is because of surgeon preference (perhaps more being seeing by neuro-otologists?). I would be curious to know more about approach selection. Where some of these possibly amenable to endonasal route but mastoid approaches chosen, or is this truly a cohort of laterally located petrous cholesterol granulomas, less so involving the petrous apex.

All of the cases were presented in our multidisciplinary skull-base team before opting for surgery. Endoscopic approaches where discussed in all of our cases but due to unfavorable anatomy for endonasal drainage a lateral approach was chosen. In retrospect – surprisingly - no endonasal route was chosen. Endoscopic and transnasal resection still remains an option in lesions located in the petrous apex in our referral centres.

We have added this nuance to the article (line 191-195).

  1. What was the recurrence rate and median follow-up time after surgery treatment?

We have discussed the recurrences and mean follow-up time elaborately in our results (see lines 165-195) and have incorporated the recurrences in figure 2, stratified per approach.

  1. How long were drains left in place postop? where did the drains exit (i.e. where did the resection cavity contents drain out into?)

The drains were left permanently in situ to ensure continual aeriation. In all approaches used the drain ends in the mastoid cavity (also after middle fossa resection).  We have added this to the article (line 159).

This manuscript is a resubmission of an earlier submission. The following is a list of the peer review reports and author responses from that submission.

Round 1

Reviewer 1 Report

This study analyzes the characteristics and evolution (with and without treatment) of 34 patients with cholesterol granuloma of the petrous apex.
The article is well organized, well designed, and correctly written and presented. The conclusions are in accordance with the results and the figures are illustrative. The main contribution is that it is a relatively large series of patients, given the rarity of this pathology, but on the other hand it does not contribute anything new to our knowledge of it, neither from the diagnostic nor the therapeutic point of view.
It can be considered a scientifically correct work, but given the lack of novelty and the fact that it is a very specialized topic, I think it would be more appropriate to publish it in a journal focused on otology or skull base pathology, but not in a cancer journal, even if the special issue is on skull base tumors. Cholesterol granuloma cannot be considered a tumor.